

# Training citizen scientists through an online game developed for data quality control

Barbara Strobl[1], Simon Etter[1], H.J. (Ilja) van Meerveld[1], Jan Seibert[1,2]

[1]Department of Geography, University of Zurich, Zurich, 8057, Switzerland

[2]Department of Aquatic Sciences and Assessment, Swedish University of Agricultural Sciences, Uppsala, SE-750 07, Sweden

*Correspondence to*: Barbara Strobl (barbara.strobl@geo.uzh.ch)

**Abstract.** Some form of training is often necessary in citizen science projects. While in many citizen science projects it is possible to keep tasks simple so that training requirements are minimal, some projects include more challenging tasks and, thus, require more extensive training. Training can hinder joining a project, and therefore most citizen science projects prefer

to keep training requirements low. However, training may be needed to ensure good data quality. In this study, we evaluated if an online game that was originally developed for data quality control in a citizen science project, can be used for training for that project. More specifically, we investigated whether the CrowdWater game can be used to train new participants on how to use the virtual staff gauge in the CrowdWater smartphone app for the collection of water level class data. Within this app, the task of placing a virtual staff gauge to start measurements at a new location has proven to be challenging; however

this is a crucial task for all subsequent measurements at this location. We analysed the performance of 52 participants in the placement of the virtual staff gauge before and after playing the online CrowdWater game as a form of training. After playing the game, the performance improved for most participants. This suggests that players learned project related tasks intuitively by observing actual gauge placements by other citizen scientists and thus acquired knowledge about how to best use the app instinctively. Interestingly, self-assessment was not a good proxy for the participants' performance or performance increase.

These results demonstrate the value of an online game for training, particularly when compared to other information materials, which are often not used extensively by citizen scientist. These findings are useful for the development of training strategies for other citizen science projects because they indicate that gamified approaches might provide valuable alternative training methods.

## 1 Introduction

Citizen science projects can be grouped into two different types with regard to data collection and training: either citizen scientists are engaged for relatively straightforward tasks so that no training is needed, or they perform more advanced tasks that require detailed instructions and training before they can participate in the project (Breuer et al., 2015; Gaddis, 2018; Reges et al., 2016). Training needs depend on the tasks within the projects and the project designers' perceived need for training. Environment-focused projects, in which citizen scientists perform simple tasks and, therefore, receive no prior

training are, for example, the global project iNaturalist, where citizen scientists take a picture of plants and animals and upload them to a server (Gaddis, 2018; Pimm et al., 2014), CrowdHydrology, where people passing by, such as hikers read the water level of staff gauges in the USA (Lowry et al., 2019), a similar water level study in Kenya (Weeser et al., 2018) or a study on the occurrence of hail in Switzerland (Barras et al., 2019). Projects, in which citizen scientists receive training prior to being able to participate, are for example CoCoRaHS, where citizen scientists operate a weather station (Reges et al., 2016), a

groundwater study in Canada, where volunteers measure the water level from wells (Little et al., 2016), a water quality study in Kenya and Germany (Breuer et al., 2015; Rufino et al., 2018) or a water clarity study in lakes in the USA (Canfield et al., 2016).

In practice, there is a range of citizen science projects and they can be positioned between these two training types, especially when the tasks are relatively easy but data quality can be significantly improved with training. An example is Galaxy Zoo,



which requires participants to classify galaxies in an online test, before they can start to submit data (Lintott et al., 2008). Another project is the Malaria Diagnosis Game, which offers a short online tutorial for players (Mavandadi et al., 2012). Some projects offer in-person training (Kremen et al., 2011; Krennert et al., 2018; Rufino et al., 2018) but for many projects training has to be online because the projects are global (e.g., CrowdWater (Seibert et al., 2019a), CoCoRaHS (Reges et al., 2016), and an invasive species training programme (Newman et al., 2010)). Computer based training can be tricky because the participants

cannot be monitored. However, Starr et al. (2014) found, that such computer training methods, e.g. via video, can be just as successful as in-person training. Computer based training, furthermore, requires less time from the project organizers once the material has been developed.

The topic of training and learning in citizen science received more interest in recent years (Bonney et al., 2016; Cronje et al., 2011; Jennett et al., 2016; Phillips et al., 2019). Bonney, et al. (2009) write "*Projects demanding high skill levels from*

*participants can be successfully developed, but they require significant participant training and support materials […].*". Many other citizen science projects that provide training, focus more on topic specific knowledge, often because this is required to complete the task successfully. Examples are the Flying Beauties project (Dem et al., 2018), the Neighbourhood Nestwatch Program (Evans et al., 2005), or invasive species projects (Crall et al., 2013; Cronje et al., 2011; Jordan et al., 2011), where participants have to learn to identify species before they can participate in the project. Some citizen science projects find that

the participants did not increase their factual learning, possibly because they were already quite advanced (Overdevest et al., 2004). Contributory projects often emphasise specific skills more than general topic knowledge. Examples of training for specific skills rather than knowledge are the Canadian groundwater study (Little et al., 2016) or the water quality study in Kenya (Rufino et al., 2018). However, "*Engagement in contributory citizen science might, by way of the methods employed, result in more data reliability but fewer science literacy gains among participants.*" (Gaddis, 2018).

A novel approach to training was developed within the CrowdWater project. The CrowdWater project explores opportunities to collect hydrological data with citizen science approaches. On the one hand, the project develops new approaches to collect hydrological data by public participation (Kampf et al., 2018; Seibert et al., 2019a, 2019b) and on the other hand the project assesses the potential value of such data for hydrological modelling (Etter et al., 2018; van Meerveld et al., 2017). In this study, the focus is on the collection of water level class observations using the virtual staff gauge approach (Seibert et al., 2019a).

This virtual staff gauge approach allows observations without physical installations, such as staff gauges (Lowry et al., 2019; Weeser et al., 2018) so that it is scalable and can be used anywhere in the world. However, it is also more challenging for the user and potentially prone to mistakes (Seibert et al., 2019a; Strobl et al., 2019). Previously we developed a web-based game for quality control of the water level class data (Strobl et al., 2019). Here, we investigate whether playing this game might also be a useful preparation for using the virtual staff gauge approach in the CrowdWater app. The objective was to evaluate whether

playing the game helped study participants to understand the virtual staff gauge approach. More specifically, we addressed the following three specific questions:

- Are participants better at placing a virtual staff gauge after they have played the game?
- Are participants better at assessing the suitability of a reference picture after they have played the game?
- Are participants more confident about their contributions after training and is this confidence related to their

75        performance in playing the game?

## 2 Background on water level class observations in CrowdWater

### 2.1 CrowdWater app

The CrowdWater smartphone app enables citizen scientists to collect data for several hydrological parameters without requiring any physical installations or equipment. The app allows the citizen scientists to set up new observation location and

to submit new observations for existing locations. The app uses OpenStreetMap (Goodchild, 2007) and thus allows geo-





referencing of observations world-wide. To start a new observation location, the citizen scientist takes a picture of a stream, showing the stream bank, a bridge pillar or any other structure that allows identification of the water level. Within the app, a virtual staff gauge is inserted onto this picture, which then becomes the reference for all further observations at this location (and is therefore called the reference picture). The virtual staff gauge is basically a sticker that is positioned as an additional

layer onto the initial picture (Fig. 1, left picture), i.e., there is no physical installation at the location. The citizen scientist can choose from three virtual staff gauges in the app, depending on the water level at the time when the picture is taken (low, medium or high; Seibert et al., 2019a). When placing the virtual staff gauge in the reference picture, the citizen scientist has to move the staff gauge so that it is level with the current water level and change the size of the staff gauge so that it covers the likely range of high and low water levels. When taking the reference picture, it is important that it is perpendicular to the

stream bank to avoid distortions when comparing the water level with the virtual staff gauge at a later time. Poor staff gauge placement is one of the most common errors. This occurs for about 10 % of the new reference pictures (Seibert et al., 2019a). The most common errors are making the virtual staff gauge too big or more rarely too small to be useful to record water level fluctuations, not placing the staff gauge on the opposite river bank, perpendicular to the flow or choosing the wrong staff gauge (Seibert et al., 2019a, 2019b). For further observations (i.e., observations at the same location at a later time), the citizen

scientist who created the reference picture or any another person looks for the structures in the reference picture (e.g., rock, bridge pillar, wall), and estimates the water level class by comparing it to the virtual staff gauge in the reference picture (Seibert et al., 2019a). This way, time series of water level class data can be obtained at each measurement locations.

**2.2 CrowdWater game**

In addition to data collection using the CrowdWater smartphone app, citizen scientists can also contribute to the project by

checking the collected water level class data in the web-based CrowdWater game (Strobl et al., 2019). The idea of the CrowdWater game is to crowdsource the quality control of the submitted water level class observations by using the pictures that were taken and submitted by the citizen scientists in the app. In the game, picture pairs are shown: the reference picture with the virtual staff gauge and a picture of the same location at a later time (Fig. 1). The task is to estimate the water level class for the later picture without the staff gauge (Fig. 1, right picture) by comparing the water level in this picture with the

reference picture, i.e. the picture with the staff gauge (Fig. 1, left picture). Citizen scientists play rounds of twelve picture pairs: eight classified pictures that have already been assigned a *"correct"* value, i.e. the median based on the evaluations of at least 15 game players and four (so far) unclassified pictures. Currently the CrowdWater game uses *"unstructured crowdsourcing"* (Silvertown et al., 2015, p. 127), which means that all votes are weighted equally to obtain the correct water level class. The order of the pictures is random, so that the player does not know whether a picture pair has already been classified or not. For

the classified picture pairs, points are obtained for choosing the *"correct"* class (6 points) or a neighbouring class (4 points) and 0 points are given if the selected class is more than one class off from the *"correct"* value). For unclassified pictures, the player receives 3 points regardless of the vote. Players can also report a picture if voting is not possible because of, for instance, unsuitable placement of the staff gauge, poor image quality, or otherwise unsuitable pictures. In this case, the player also receives 3 points. The repeated evaluations of the same pictures by multiple players provides quality control of the incoming

water level class data (Strobl et al., 2019).




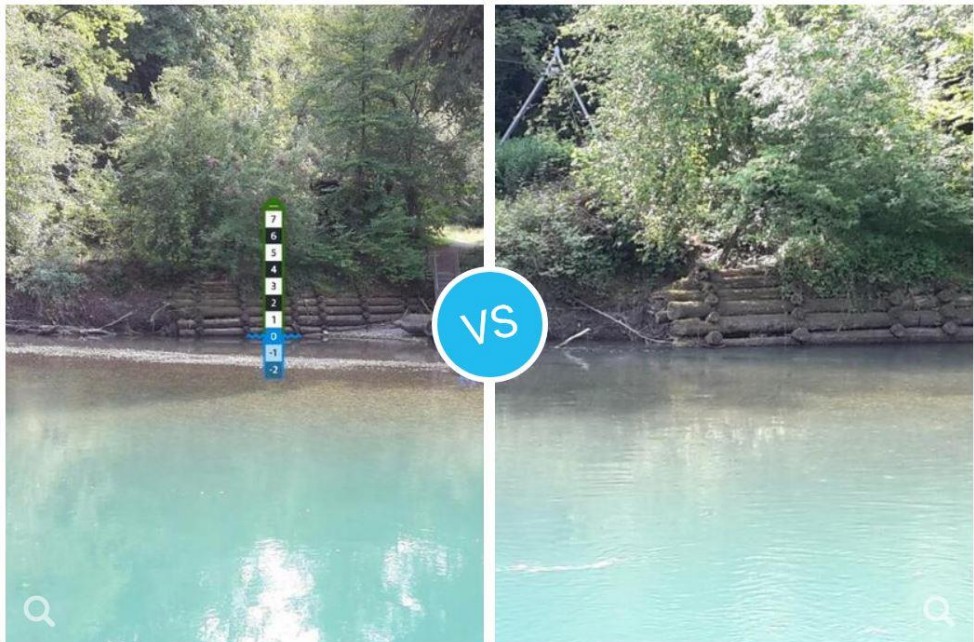

**Figure 1: An example of a reference picture with the virtual staff gauge (left) and a picture from an observation at the same location at a later time (right). The logs of the streambank can be used as a reference to estimate the water level class.**

**2.3 Previous CrowdWater training and motivation for this study**

When using the CrowdWater app, citizen scientists take a picture of the observation location and upload it, similar to iNaturalist (Gaddis, 2018; Pimm et al., 2014) or iSpot (Silvertown et al., 2015). When starting observations at a new location, some interpretation is needed, which requires an understanding of the possible range of water levels and determination of the current water level. The data collection protocol is, however, simpler than for many projects that do require training, therefore a low-intensity training seems to be advisable for the CrowdWater project.

As a first step, manuals (https://www.crowdwater.ch/en/crowdwaterapp-en/) and instruction videos (https://www.youtube.com/channel/UC088v9paXZyJ9TcRFh7oNYg) were provided online, but in our experience (and based on the number of watches on YouTube) these are not frequently used. Thus some citizen scientists occasionally still make mistakes when submitting data in the CrowdWater app, primarily when starting a new location for observations and placing a virtual staff gauge onto the reference picture (Seibert et al., 2019a). Our first approach to handle these mistakes was to

implement a method of quality control, in order to either filter out or correct erroneous submissions. This quality control method was gamified in the CrowdWater game. The CrowdWater game proved successful in improving the quality of the water level data submitted through the app (Strobl et al., 2019). Shortly after launching the game, we received anecdotal evidence, such as direct feedback from players that the game also helped them to better place staff gauges and to better estimate water level classes. This feedback was confirmed through a short survey sent out to CrowdWater game players for a different

study (Strobl et al., 2019). Roughly a quarter of all players at the time filled in the survey (36 players). When asked if playing the game helped them to be more aware of how to place a staff gauge in the app, 79 % agreed. Furthermore, 58 % of players agreed that the game helped them to better estimate water level classes in the app. The other players indicated no change in their abilities and none of the survey players indicated a deterioration of his or her skills. Essentially, the players are training each other in the game, as the correct votes are provided by other players and the score per picture pair shows the new player

if (s)he is correct or not. This is similar to iSpot, where experts train beginners in species recognition (Silvertown et al., 2015).



Through the CrowdWater game, players learn which staff gauges are difficult to read and which ones allow easy comparison of the water levels (Strobl et al., 2019).

This motivated us to investigate if the CrowdWater game can be used to train potential citizen scientists to place the virtual staff gauge in the CrowdWater app correctly. It is better to train citizen scientists before participation, so that they provide 145 useful data, rather than filtering data from untrained citizen scientists afterwards. Filtering wrong data afterwards wastes the time of the citizen scientists and erroneous data can be missed by the filter. In the CrowdWater project, it is particularly important to place the virtual staff gauge correctly because all subsequent observations at an observation location are based on this virtual staff gauge (i.e., a poorly placed staff gauge will influence all following observations).

The CrowdWater game is a project specific training tool, meant to improve the reliability of CrowdWater app observations 150 and does not aim at improving scientific literacy. This is similar to some other citizen science projects, especially contributory projects where data is crowdsourced (Crall et al., 2013). Improving the hydrological knowledge was not necessary in our case, as the data can easily be collected without such background knowledge. However, other materials that provide such knowledge and a link to an open massive online course are provided on the project website.

## 3 Methods

### 3.1 Training study

The aim of this study was to assess if the CrowdWater game can be used to train new participants to correctly place the virtual staff gauge in the CrowdWater app. The placement of the staff gauge is the most important metric for this study because this is the most crucial task for CrowdWater app users starting a new observation location. Rating reference pictures gave additional insight into whether participants can recognise well and poorly placed staff gauges, regardless of whether or not they can place 160 them well themselves.

The training study consisted of a number of tasks that were executed before and after playing the game. To focus on the research questions and to exclude other factors, such as differences between locations, flow conditions or daylight, the study was mainly conducted indoors at a computer. In addition, basic demographic information was collected. For each participant the experiment took 60-90 minutes. All instructions and questions were formulated in English; all participants had a good 165 command of English. The study was conducted between August and October 2018, apart from a small outdoor task, which was completed by the participants at a later time. The full study can be found in the supplementary material: Training study.

### 3.1.1 Study tasks

The six tasks of the training study can be divided into tasks before the training, the training, and tasks after the training (Fig. 2). These tasks were completed by each participant in the same order.

Tasks before the training:

- **First task (staff gauge placement)**: The study participant looked at 18 stream pictures of the river Glatt (supplementary material: Stream pictures). All pictures show the same location but were taken from different angles and perspectives. Some were well suited for placing a virtual staff gauge, others were moderately suitable and some were not suitable at all. Without receiving any further information, the participants were asked to choose one of the 175 18 pictures and to place a virtual staff gauge onto the picture. This was done using an interface on the computer that looked similar to that in the CrowdWater app.

- **Second task (rating of reference pictures)**: The participant looked at 30 different reference pictures (for examples see supplementary material: Examples of reference pictures for the rating task). These pictures were chosen from the reference pictures that were uploaded by citizen scientists using the CrowdWater app. The pictures were selected to





represent a range of well, moderately, and poorly placed virtual staff gauges. The participants rated each of the 30 reference pictures as *"unsuitable"*, *"rather unsuitable"*, *"rather suitable"* or *"suitable"*.

Training task:

- **Third task (game)**: The participant played an adapted version of the CrowdWater game. In this version, the participant estimated the water level class of 50 picture pairs. The regular CrowdWater game only offers twelve
picture pairs per day, so this extended version corresponds to the training effect of about four rounds of the game. Participants did not receive any explanation on the game but could use the help button to obtain more information on the game.

Tasks after the training:

- **Fourth task (staff gauge placement)**: The participant repeated the first task and was asked to place the virtual staff
gauge again for the river Glatt. The participant received the same 18 pictures but was free to choose another picture and to place the virtual staff gauge in a different location, angle or size compared to the first task, or to choose the same picture and to place the staff gauge similarly.
- **Fifth task (rating of reference pictures)**: The participant repeated the second task for a different set of 30 reference pictures from the app. The distribution of well, moderately and poorly placed virtual staff gauges was roughly the
same as in the second task.
- **Sixth task (staff gauge placement)**: The participant used the CrowdWater app (instead of the online interface used for the earlier tasks) to create and upload a reference picture for a stream of their choice. Initially the task was meant to be completed within two weeks after completing the first five tasks, however, not every participant completed the task within this timeframe (at the latest by March 2019) and six participants did not complete this task at all.

After placing the staff gauge online (first and fourth task) and rating the reference pictures (second and fifth task) participants answered several questions to assess the difficulty of the task, their own performance, and their confidence in completing these tasks correctly. After the training (third task), participants were asked about the difficulty of the game and whether they thought the game was fun.





| | Nr. | Task | Score name | Max. score | Good score |
|---|---|---|---|---|---|
| **BEFORE TRAINING** | 1. | | Placement score | 13 points | ≥ 10 points |
| | 2. | 30x | Rating score | 90 points | ≥ 75 points |
| **TRAINING** | 3. | 50x | Game score | 300 points | ≥ 245 points |
| **AFTER TRAINING** | 4. | | Placement score | 13 points | ≥ 10 points |
| | 5. | 30x | Rating score | 90 points | ≥ 75 points |
| | 6. | | Placement score | 13 points | ≥ 10 points |

**Figure 2: Schematic overview of the study tasks, divided into before training, training, and after training tasks. For each task the maximum number of points and the (chosen) value for a good performance are given.**

### 3.1.2 Assessment of the different tasks

The performance of the participants for the different tasks was evaluated based on a score. The scores before and after playing the game (i.e., the training) were compared to determine the effect of playing the game. The scoring system was determined prior to the start of the study according to assessment criteria that were based on previous experiences with pictures submitted through the app and expert judgement (by Strobl and Etter). A separation of the individual scores into *"good"* and *"poor"* was, while somewhat arbitrary, necessary to be able to distinguish the effects the training on the participants who needed it most, i.e. those who had poor scores before the training.

For the staff gauge placement task (first, fourth and sixth task) points were given for five different placement criteria. The maximum placement score was 13. A placement score of 10 or higher was considered good because these reference pictures can still be used and would have been left in the CrowdWater database if they were submitted through the app (Fig. 3):

1. *Perspective of the picture*: The 18 pictures of the river Glatt were taken from different angles and perspectives and assigned a score: 0 (unsuitable), 1 (rather unsuitable), 2 (rather suitable) and 3 (suitable). The participant could gain more points for the choice of the picture than the other criteria for placing a staff gauge because this is essential for a good reference picture. Because every participant fulfilled the outdoor task (sixth task) for a different stream and the participants could choose a location themselves, points could not be assigned a priori. However, the location and the picture frame were assessed and a score between 0 and 3 was given based on expert judgement (by Strobl and Etter).

2. *Choice of the staff gauge*: Participants can choose from three different virtual staff gauges depending on the water level at the time that the picture was taken (low, medium or high). The staff gauge for low flow was considered correct, as the water level was low at the time that the 18 pictures of the Glatt were taken. The score for the selected staff gauge varied between 2 (staff gauge for low flow), 1 (staff gauge for medium flow) and 0 (staff gauge for high





flow). For the outdoor task with the app (sixth task), the situation was assessed based on the water level and points were assigned for the correct assessment of low, medium or high flow by the participant.

3. *Location of the staff gauge*: If the staff gauge was placed on the opposite stream bank, as it should be, 2 points were given; if the staff gauge was incorrectly placed on the participant's side of the stream or in the middle of the stream, 0 points were given.

4. *Angle of the staff gauge*: The staff gauge should be placed perpendicular to the flow in the stream to avoid contortions of the perspective for future water level estimates. If the staff gauge was placed perpendicular to the flow (± 10°), 2 points were given. If the angle was less than 45°, 1 point was given, and if it was larger than 45°, 0 points were given.

5. *Water level mark*: The blue wave of the staff gauge should be located at the actual water surface of the reference picture. If this was the case 2 points were given, if the blue wave was only slightly off, e.g., due to reflections on the water surface, 1 point was given and if the blue wave was not placed at the water surface 0 points.

Very rarely, two virtual staff gauges were placed in the reference picture (twice before the training (first task), once after the training online (fourth task) and once after the training in the app (sixth task)). We assume that this was most likely due to technical difficulties. In these cases, we subtracted 1 point from the participant's score. This, however, had hardly any effect on the results.





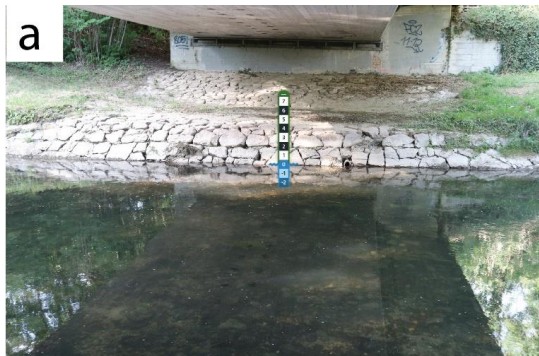

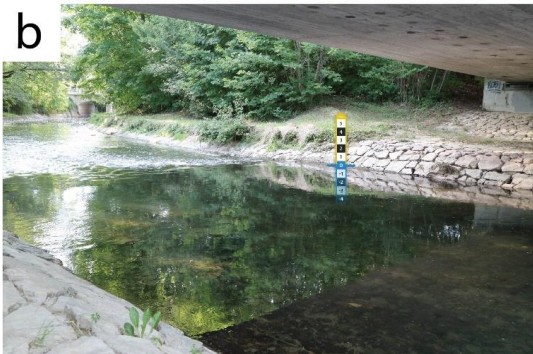

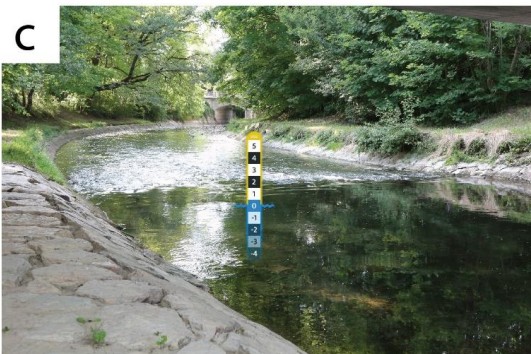

**Figure 3: Examples of staff gauge placements: (a) 13 points, i.e., a full score, (b) 10 points, just enough points to still be considered suitable for future contributions in the app, and (c) 3 points, the lowest score obtained throughout the study.**

The rating of the reference pictures (second and fifth task) was evaluated using a rating score. The participant's choice between *"unsuitable"*, *"rather unsuitable"*, *"rather suitable"* and *"suitable"* was compared to the expert judgement of the reference pictures (by Strobl and Etter). If the participant picked the same suitability class as the experts, 3 points were given. For each class deviation from the expert judgement, one point was subtracted. Thus the maximum score was 90 points (30 reference pictures times 3 points per picture). A score of 75, which corresponds to being five times one class and five times two classes off, and choosing the correct class 20 times, was still considered good.

For the training task (the fifth task, the game), the participants received points for each picture pair that they compared. Similar to the actual CrowdWater game, they received 6 points if they chose the *"correct"* class, i.e. the median of the votes of all previous CrowdWater game players, 4 points if they chose a water level class that was one class away, and 0 points if they chose a class that was more than one class away from the median. When reporting a picture pair, the participant received 3

points. The maximum score for the training task was 300 points (a maximum of six points for each of the 50 picture pairs). The threshold for a good game score was set at 245 points, which reflects a situation where a participant chose the correct class





for 35 out of the 50 picture pairs, was one class off five times, more than one class off for another five picture pairs, and reported five pictures (we considered five pictures unsuitable and would thus have reported them).

### 3.1.3 Data analysis

The scores for the tasks before and after the training were compared for each participant using two paired statistical tests: the paired sample t-test for normally distributed data and the Wilcoxon test for not normally distributed data (Table 1). We used a one-sided test to test determine if the difference in the scores before and after the training was larger than zero and a two-sided test to determine the significance of the difference in the scores between the computer based and app based staff gauge placement (i.e. between the fourth and the sixth task). We used a significance level of 0.05 for all tests. We performed the tests

for all participants together but also divided the participants based on their placement score before the training (first task) in order to determine the effect of training for people who initially did not install the virtual staff gauge correctly. In order to see whether the game performance was correlated to placement or rating score improvements, we also split the data based on the game score to assess if participants with a high game score improved their performance after the training even more than participants with a low game score,. We used Spearman rank correlation ($r_s$) to evaluate the relation between performance (i.e.,

scores) and confidence of the participants in their performance, as well as between the performance and the stated difficulty and fun rating.

Table 1: The statistical tests were chosen based on whether or not the data were normally distributed according to the Shapiro-Wilk test. The test for the placement score compared before and after training scores, as well as after training and app scores. The test for the rating score was used to determine the significance of the difference in the before and after training rating scores.

| Data | Data subset | Results of the Shapiro-Wilk test | Statistical test of the training effect |
|---|---|---|---|
| Placement score | All participants | Not normally distributed | Wilcoxon test |
| | Participants with a low placement score before the training | Not normally distributed | Wilcoxon test |
| | Participants with a good game score | Not normally distributed | Wilcoxon test |
| | Participants with a bad game score | Not normally distributed | Wilcoxon test |
| Rating score | All participants | Normally distributed | Paired sample t-test |
| | Participants with a low rating score before the training | Not normally distributed | Wilcoxon test |
| | Participants with a good game score | Not normally distributed | Wilcoxon test |
| | Participants with a low game score | Normally distributed | Paired sample t-test |


### 3.2 Study participants

The participants for this study were recruited through various channels. The University of Zurich offers a database with potential study participants in the vicinity of Zurich; people in this database were contacted via email. Additional emails were sent out to staff and students of the Department of Geography. Friends, colleagues and family helped to recruit participants

from their social network as well. Local study participants could complete the online part of the study in a computer room at





the University of Zurich at specified times; all other participants received the link and completed the study on their own. All participants completed the first five tasks individually in one session.

The participants in this study had not previously used the CrowdWater app, nor played the CrowdWater game. In total, 52 participants completed the first five tasks of the study. Six out of the 52 participants did not complete the outdoor app task

afterwards, but their results were included in the analyses as far as possible. Of the 52 participants, 32 (62 %) were female and 20 (38 %) were male. Age data were collected in age groups: 6 % of the participants were under 20 years old, 79 % of the participants were 21-40 years old, 8 % were 41-60 years old and 8 % were 61-80 years old. The highest education was secondary school for 4 % of the participants, high school for 12 % of the participants, university (BSc/MSc or similar) for 79 % of the participants and a PhD for 6 % of the participants. This higher education level than the Swiss average and large group

of young people (< 40 years) is due to the recruitment of the participants at the University of Zurich. The level of education of the CrowdWater citizen scientists is unknown but 89 % of the 36 CrowdWater game players, who filled in a survey about the game were university educated and 75 % were under the age of 40 (Strobl et al., 2019). For a survey about the motivations of CrowdWater app users, as well as citizen scientists from a different phenological citizen science project (Nature's Calendar ZAMG), 66 % of the respondents were university educated and 51 % were under the age of 40 (Etter et al., in review).

**4 Results**

**4.1 Training results**

Almost two thirds (62 %) of study participants had a good score (≥ 245 points) for the game. The highest game score was 274, the average score was 248. The lowest score (160 points) was an outlier, the second lowest was 211 points (Fig. 4). Interestingly the participant with the lowest game score found the game *"rather difficult"*, but nonetheless still *"a bit of fun"*, adding *"It*

[the game] *was quite tricky. I was curious if my answer is right or wrong"*.

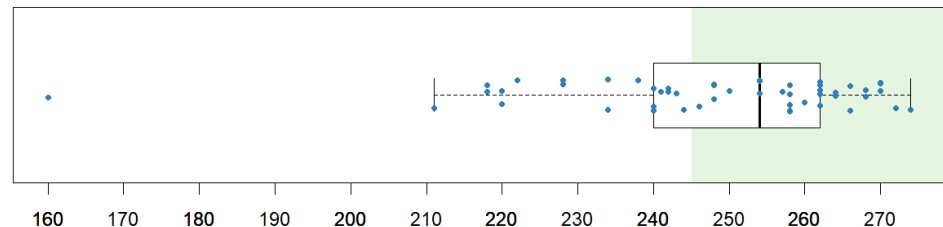

**Figure 4: Game score for each study participant. Scores ≥ 245 points are considered good (indicated by the green background). The box represents the 25th and 75th percentile, the line the median, the whiskers extend to 1.5 times the interquartile range. The individual scores (blue dots) are jittered to improve the visibility of all points.**

In the game, participants can report a picture pair if they think that it is not possible to vote on a water level class. The reason for reporting a picture pair can be selected from a drop down menu. The report function was used by 16 participants (31 %). It is unknown if the other 36 participants did not find the report function or if they did not think it was necessary to report any of the picture pairs. Most of the participants who used the report function, reported between one and six picture pairs. Although, one participant reported ten and another participant reported twelve picture pairs. Out of the 50 picture pairs in the game, 22

were reported at least once and one picture pair was reported seven times. When choosing the 50 picture pairs for the game, we included five picture pairs that should be reported (Fig. 5). In other words, there were 57 reports in total, 38 of which were not valid (i.e., our expert knowledge suggests that the picture pairs could be used to determine the water level class). For some of these cases, participants considered a spot unsuitable because they did not realise that they could see the entire picture if they clicked on it and therefore thought the reference picture did not have a staff gauge. In another case, they may have been

confused by a slightly different angle in the picture for the new observation. The most common reason for reporting a picture





was *"The location has changed and the reference image is unrecognizable"*. This was indeed a problem with some of the picture pairs (Fig. 5).

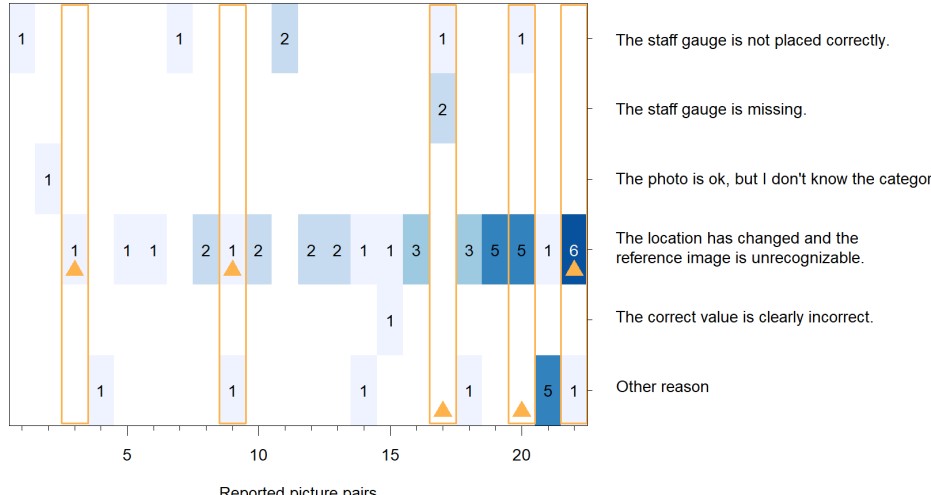

**Figure 5: The number of times that a picture pair was reported and the reason for reporting the picture pair (y-axis) for the 22**
**picture pairs in the game that were reported at least once (x-axis). The picture pairs that should have been reported based on expert assessment prior to the training study are framed with an orange rectangle, the orange triangle indicates the reason based on expert assessment. The blue shading represents the number of reports per picture pair (as also indicated by the printed number).**

**4.2 Staff gauge placement**

**4.2.1 Placement scores before training**

The staff gauge placement score before the training (first task) was 10 or higher for 70 % of the participants, i.e., the majority of the participants placed the staff gauge in a way that is suitable for further observations. This is a good performance considering that the participants did not receive any training yet. Training is more important for the 30 % of participants who had a low score before the training. The lowest scores were two points (one participant) and three points (two participants).

**4.2.2 Placement scores after training**

The placement scores generally improved after the training and were statistically significantly different from the scores before the training (Wilcoxon test, $p < 0.01$; Fig. 7). Improvement is especially important for the participants who had a low placement score before the training, therefore, the participants with a low score ($< 10$ points) were assessed separately. For this group the median placement score improved significantly with training as well (Wilcoxon test, $p < 0.01$). Of the 16 participants with a poor placement score before the training, ten improved their staff gauge placement sufficiently to make it useful for future
observations. Participants who performed well before the training, have less possibility to improve the placement, and also need to improve their staff gauge placement score less. However, for two of the participants with a good score before the training, the score was poor after the training (Fig. 6 and 7). The placement score improved for participants with a good game score (Wilcoxon test, $p < 0.01$), but not for participants with a low game score (Wilcoxon test, $p = 0.11$).





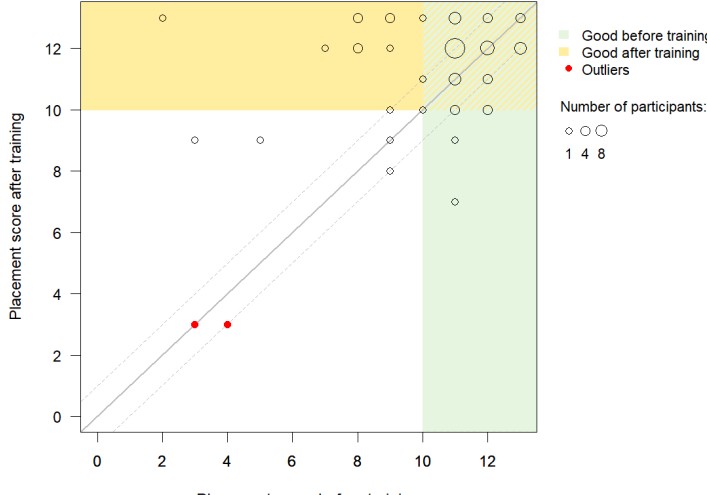

**Figure 6: Placement scores before the training (x-axis) and after the training (y-axis). The circle size indicates the number of participants with the same scores. The green background indicates participants who already performed well in placing the staff gauge before the training (score ≥ 10) and the yellow background indicates participants who performed well after the training (score ≥ 10). The solid grey line indicates the 1:1 line (i.e., the same score before and after the training), while the dashed lines indicate a difference of only one point. Points in the upper left triangle indicate an improvement in staff gauge placement after the training.**
**The red circles indicate outliers.**



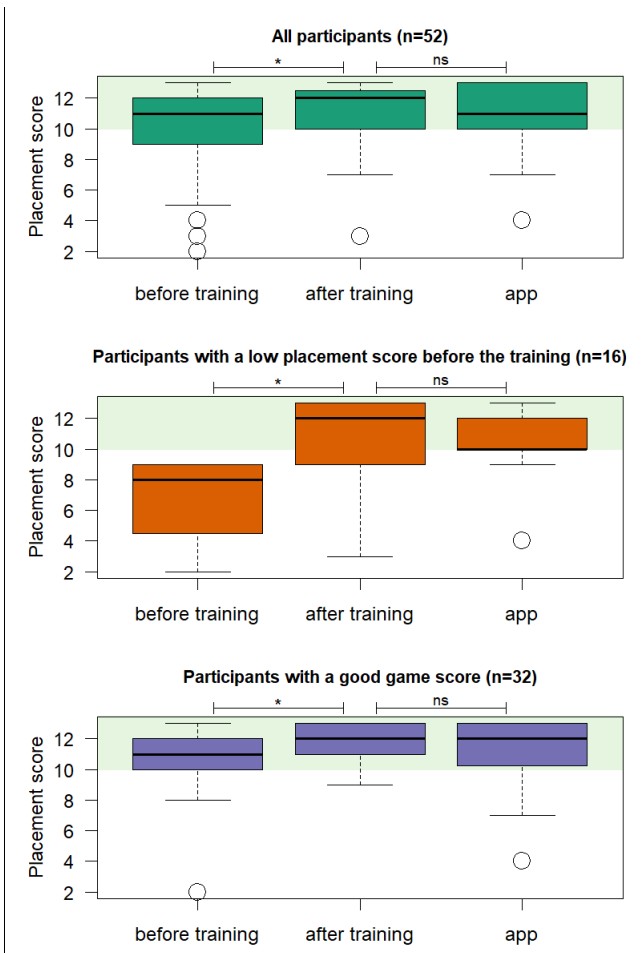

**Figure 7: Box plots of the placement scores before the training (first task), after the training online (fourth task) and outdoors with the app (sixth task) for all participants (upper plot), for participants who had a low placement score before the training (middle plot) and for participants who had a good game score (lower plot). There was a statistically significant difference in the placement**

**scores before and after the training for all groups (indicated with the \*) and no statistically significant difference between the computer based task and the outdoor app task (indicated with "ns") after the training based on the Wilcoxon test (p < 0.05). The green shading indicates a good score.**

Nineteen participants (37 %) picked a different picture for the staff gauge placement after the training. Eight of these participants chose a stream picture with the same suitability score as the first one, nine selected a better stream picture, and

two chose a picture that was worse than their original choice. The other 33 participants chose the same stream picture as before, however the good scores even before the training suggest that most of them also did not need to change the picture. The participants who changed the stream picture had a median placement score of 9 before the training and 12 after the training. The participants who chose the same stream picture, had a median placement score of 11 before the training and of 12 after the training. Before the training, 37 participants chose a reference picture with a score of 3, nine with a score of 2, five with a

score of 1 and only one participant chose a reference picture with a score of 0. Of the six participants who had a score of 0 or 1 before the training, four had a score of 2 or 3 after the training. For two participants the reference picture score remained 1. Except for one participant, all participants who performed well in the training task (game score ≥ 245 points) had a good placement score (≥ 10) after the training. However, the opposite was not the case: participants with a low game score (< 225 points) sometimes still improved their placement score after the training and all had a good placement score (≥ 10) after the

training (Fig. 8). The participant with the biggest improvement in staff gauge placement (from 2 points to 13 points) had an



excellent game score of 262 points (Fig. 8). Participants who obtained a low score for the staff gauge placement after the training all had an average score in the game (228-243), except for one with a high game score (248; Fig. 8).

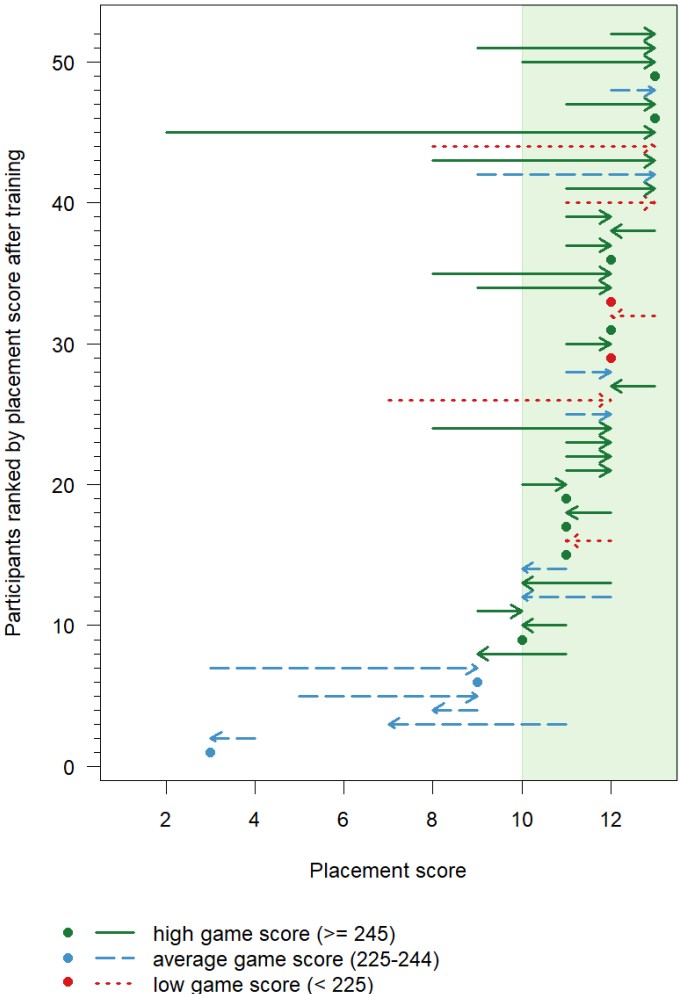

**Figure 8: Placement scores before and after the training (x-axis); arrow points from before to after training score, dots indicate no**
**change in the placement score) per participant (y-axis) coloured according to the game score they obtained during the training.**

There was no statistically significant difference between placement scores after the training for the online (fourth task) and the outdoor task with the app (sixth task). The results were not statistically significantly different for all participants (p = 0.50), for participants with a low placement score before the training (p = 1.00), nor for participants with a good game score (p = 0.20) or for participants with a bad game score (p = 0.57, Fig. 7). This indicates that the online task can be used as a proxy for
handling the app.

**4.2.3 Placement score outliers**

When plotting the placement score before the training and after the training, two outliers were visually identified (Fig. 8). Both participants had a low score before the training but unlike other participants also a low score after the training. These two participants received few points across all assessment criteria for staff gauge placement and also had a below average game





score (242 and 228 points). They rated the game as *"rather difficult"* and *"very difficult"* and when asked whether they enjoyed playing the game stated *"neutral"* and *"It wasn't fun at all"*. Surprisingly both participants were confident that the reference picture for the staff gauge placement was *"rather suitable"*. Both participants changed their impression of the difficulty of the staff gauge placement (first task) from *"very easy"* before the training to *"rather easy"* and *"neutral"* after the training (fourth task).

### 385    4.3 Rating of reference pictures

#### 4.3.1 Rating scores before the training

Even though the majority of the participants recieved already a good staff gauge placement score before the training, only 13 % of the participants had a good rating score (≥ 75) before the training. Only 9 % of the participants had a good score for both staff gauge placement and rating before the training. The highest rating score before the training was 80 and the lowest score 390    54; the average score was 68 points.

#### 4.3.2 Rating scores after the training

The rating scores improved after the training (Fig. 9 and 10). The median difference in the rating score before and after the training was statistically significant larger than zero, for all participants (paired-sample t-test, $p < 0.001$), for participants with a low rating score before the training (Wilcoxon test, $p < 0.001$), for participants with a good game score (Wilcoxon test, $p <$ 395    0.001), and for participants with a low game score (paired-sample t-test, $p = 0.02$; Fig. 10).

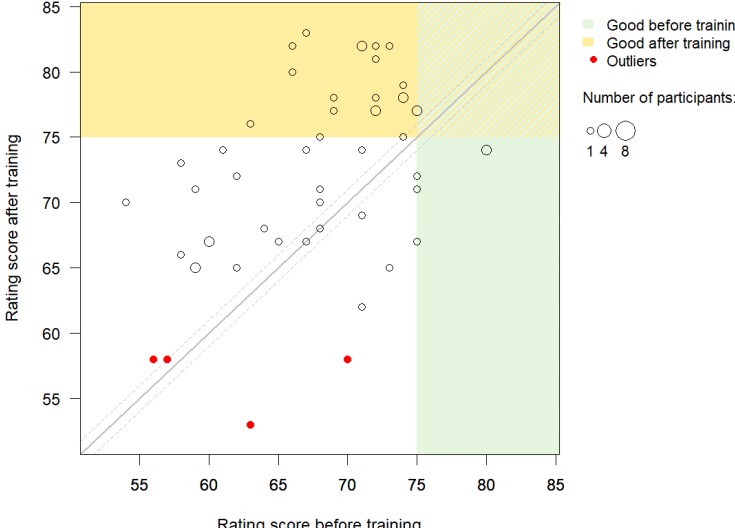

**Figure 9: Rating scores before the training (x-axis) and after the training (y-axis). The circle size indicates the number of participants with the same scores. The green background indicates participants who already performed well (score ≥ 75) before the training and the yellow background indicates participants who performed well after the training (score ≥ 75). The solid grey line indicates the 1:1** 400    **line (i.e., the same score before and after the training), while the dashed lines indicate a difference of only one point. Points in the upper left triangle indicate an improvement in the rating score after the training. The red circles indicate outliers.**




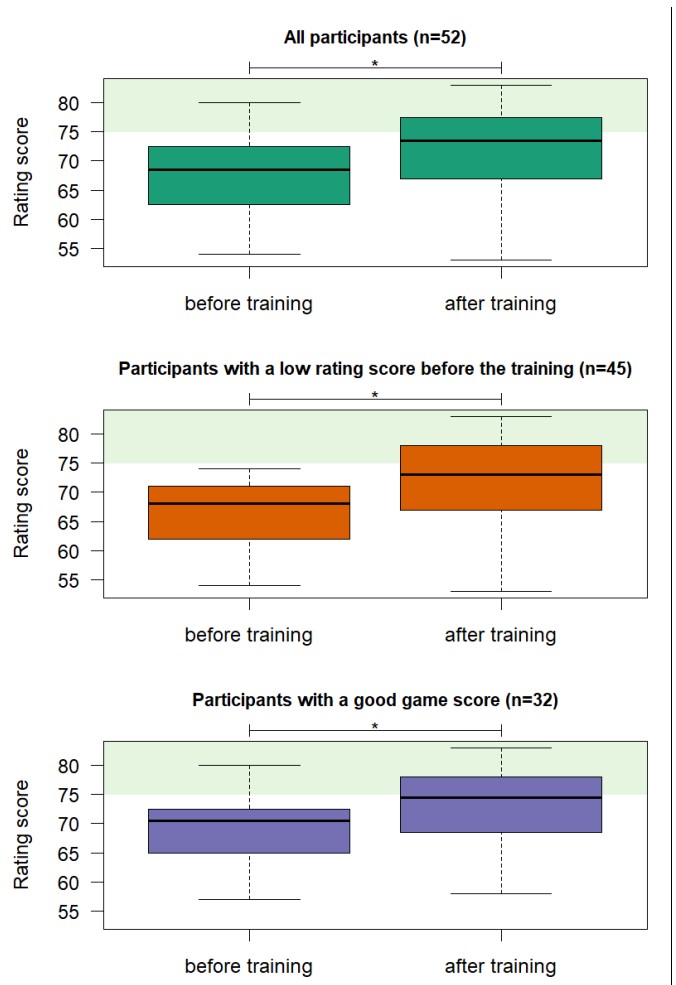

**Figure 10: Boxplots of the rating score before and after the training for all participants (upper plot), for participants who had a low rating score before the training (middle plot) and for participants who had a good game score (lower plot). The difference was statistically significant for all groups based on the Wilcoxon test (p-value < 0.05, indicated with the \*). The green shading indicates a good score.**

The rating scores can also be analysed per picture. A single picture can receive between 156 points (all 52 participants chose the correct suitability class and received 3 points) and 0 points (all participants chose the suitability class that is furthest from the correct class). The score was higher for the reference pictures that were determined to be *"unsuitable"* by the experts before the study (median: 139; range 77-152) than for the pictures that the experts rated as *"suitable"*, *"rather suitable"* and *"rather unsuitable"* pictures (median: 120-121; Table 2). This indicates that participants were better at identifying the *"unsuitable"* pictures than the more suitable pictures (Table 2).



**Table 2: Number of pictures to be rated before and after the training per suitability category (as determined prior to the study by the experts) and the median, average and range in rating scores for the pictures in each category. Each picture can receive a maximum rating of 156 points if all 52 participants chose the correct category and therefore gained three points.**

| Suitability category | Number of pictures | | Rating score (0-156) | | |
|---|---|---|---|---|---|
| | Before training (second task) | After training (fifth task) | Median | Average | Range |
| **Unsuitable** | 8 | 8 | 139 | 133 | 77-152 |
| **Rather unsuitable** | 4 | 3 | 121 | 118 | 105-128 |
| **Rather suitable** | 6 | 6 | 121 | 119 | 96-132 |
| **Suitable** | 12 | 8 | 120 | 116 | 66-138 |

### 4.3.3 Rating score outliers

Outliers for the rating scores were less obvious than for the placement scores, although there appear to be four outliers (Fig. 9, red circles). One participant was also an outlier for the staff gauge placement. The game scores and the assessment of difficulty and fun of the game varied for these four participants. The confidence in their own performance when rating the reference pictures was mixed before the training, but never lower than *"neutral"*. After the training, all four participants were confident about their performance and found the task either *"rather easy"* or *"very easy"*.

### 4.4 Confidence, difficulty and fun

#### 4.4.1 Confidence and difficulty in staff gauge placement and rating the reference pictures

The participants were in general quite confident in their performance and their confidence increased after the training (from 67 % to 98 % of participants for staff gauge placement and from 62 % to 90 % for rating the reference pictures; Fig. 11). As shown above for the outliers in the placement score and rating score, the participants' confidence in their performance was not correlated with their actual performance, neither before nor after the training ($|r_s| \leq 0.23$, $p \geq 0.11$).

Before the training, participants thought that the placement of the staff gauge was a relatively easy task but the level of difficulty was roughly equally split between *"difficult"*, *"neutral"* and *"easy"* for the rating of the reference pictures (Fig. 11, lower row). Participants generally considered the tasks easier after the training (72 % of the participants said that the placement of the staff gauge was easy before the training vs. 84 % of the participants after the training;  43 % of the participants thought that rating the reference pictures was easy before the training vs. 71 % after the training). Similar to the results for confidence, the assessment of the difficulty of the task was not directly related to the performance, neither before nor after the training ($|r_s| \leq 0.16$, $p \geq 0.30$).



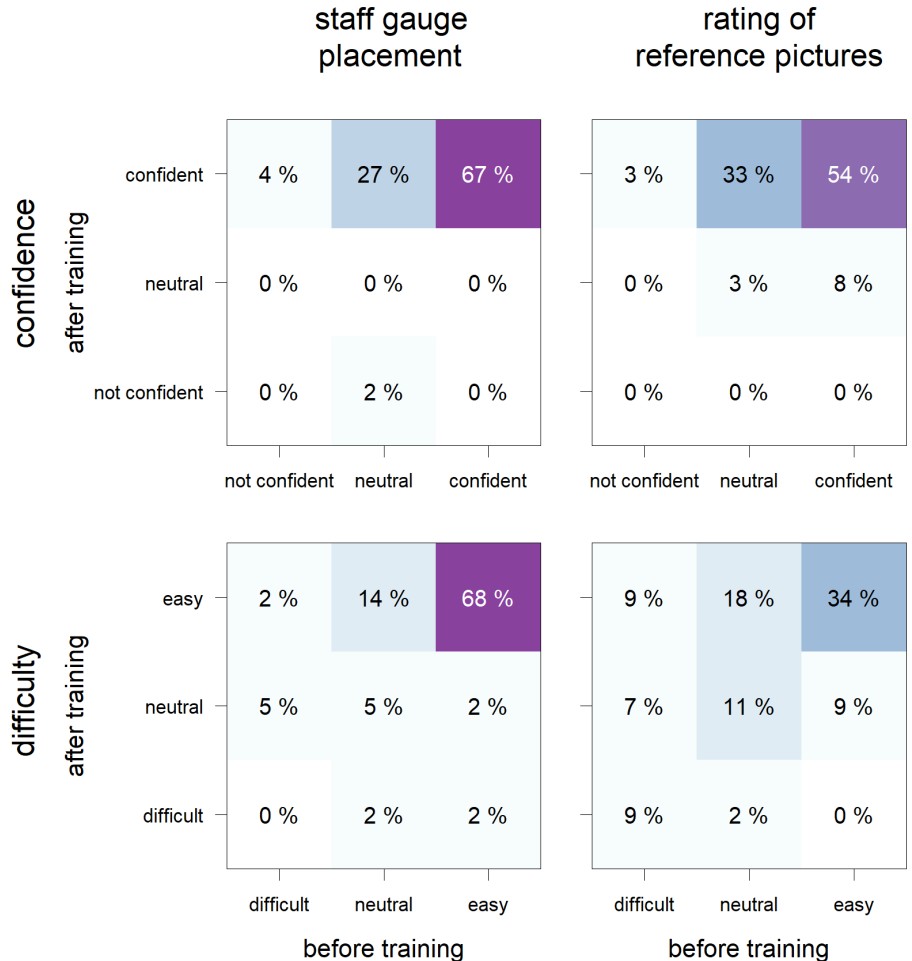

**Figure 11: Percentage of participants who chose a certain confidence level (first row) and their assessment of the difficulty of the task (second row) for staff gauge placement (first column) and rating of reference pictures (second column) before the training (x-axis) and after the training (y-axis). Darker colours indicate that a high percentage of participants chose these options.**

### 4.4.2 Difficulty and fun of the game

Two thirds of participants thought that playing the game was fun, but when rating the difficulty they were almost equally split between *"difficult"*, *"neutral"* and *"easy"* (Fig. 12). All participants who thought that the game was not fun (21 %), thought that the game was either difficult or neutral. The level of fun and difficulty was correlated ($r_s = 0.43$, $p < 0.01$). Nonetheless, 11 % of the participants stated that they had fun during the game, but also thought it was difficult.



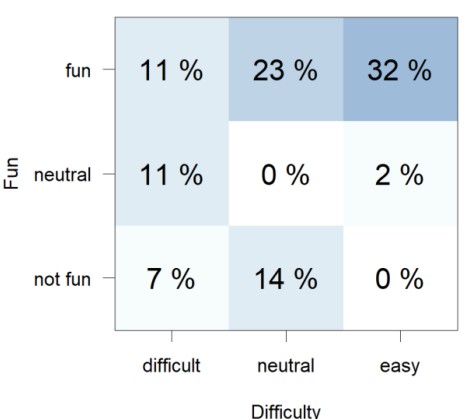


**Figure 12: Percentage of participants who chose a certain category for the difficulty (x-axis) and the fun (y-axis) for the game. Darker colours indicate that more participants chose these options.**

### 4.5 Feedback

Participants had the option to provide unstructured feedback at the end of the online study (after the fifth task); 15 participants

decided to do so. Five participants mentioned various issues that had been unclear to them during the study, four commented that they had enjoyed taking part in the study, two specifically mentioned that they thought that the training had helped but one participant stated that s(he) thought the training had not helped. Two participants stated that they thought the study was difficult and two gave feedback on the technical implementation of the study.

## 5 Discussion

**5.1 Does the CrowdWater game help participants to place a staff gauge suitably?**

The staff gauge approach was developed as an intuitive approach to collect water level data, so that many citizen scientists would be able to contribute. Such a simple approach is often recommended to citizen science project initiators (Aceves-Bueno et al., 2017). Many other citizen science projects, such as CrowdHydrology and iNaturalist, also chose a data collection method that is deliberately kept easy so that citizen scientists do not require training in advance (Gaddis, 2018; Lowry et al., 2019).

When starting a CrowdWater station for water level class observations, the most difficult task is placing the staff gauge. This is also the first thing that most citizen scientists who start using the app do. All follow-up observations are much easier to record in the app. However, the staff gauge placement is an essential task, as all subsequent observations of water level classes are based on the reference picture. This is not ideal, as the citizen scientist might not have fully understood the concept of the virtual staff gauge yet when making the first observation. Mistakes in the placement of the virtual staff gauge occur in about

10 % of the cases.

In this study, most participants (70 %) were already good at placing a staff gauge, even before receiving any training. This indicates that the virtual staff gauge is indeed intuitive to use. Training is especially important for the participants who did not place the staff gauge well before the training, i.e., citizen scientists who do not intuitively understand how to place the staff gauge in the app. Starr et al. (2014) reached a similar conclusion in a study that compared different training methods for plant

identification and also focused on the beginner group, as this allowed to clearly see the training effects. Whilst the CrowdWater app is reasonably intuitive, the fact that we do sometimes receive app submissions with mistakes (Seibert et al., 2019a) suggests that training could be beneficial. The mistakes made when using the app closely resemble the mistakes made by participants in this study and included making the staff gauge too big, not placing the zero line on the water level, or choosing a picture with an angle that distorts the image and hampers further observations at this location. Playing the CrowdWater game can help

to avoid these mistakes in a playful manner for some of the participants (63 % of the participants who performed poorly prior





to training did well after training). Based on these findings, we will suggest that new citizen scientists play the CrowdWater game before setting up a new measurement location.

Playing the CrowdWater game was not helpful for all participants; some participants who had a low placement score before the training still had a low placement score after the training as well. Rinderer et al. (2015) reported a similar case, where some

groups did improve their skills at classifying soil moisture but others did not. In the context of this study, this might be due to the CrowdWater game being an implicit approach to training, instead of an explicit one. At no point during the study did we provide theory about staff gauge placement nor did we mention the essential criteria of a good virtual staff gauge placement to participants (e.g. angle, size, placement on water level). Most participants intuitively understood this after playing the game because they noticed that a poor placement of the staff gauge made estimation of water level classes for subsequent

observations more difficult. The benefit of such an implicit approach is that it is likely more fun than merely providing the theory (which is given on the CrowdWater homepage and explained in instruction videos). Nonetheless, some participants might have preferred explicit, written instructions on what to look for, instead of having to acquire this knowledge themselves. We would, therefore, recommend that citizen science projects offer theoretical material in addition to a gamified training approach. Newman et al. (2010) encourage citizen science project leaders to provide many different training approaches to

accommodate different learning types.

When rating the reference pictures, participants were better at recognising unsuitable pictures, compared to rather unsuitable, rather suitable or suitable reference pictures. The boundaries between the intermediate categories are of course vague and somewhat subjective but it is very encouraging that participants could accurately identify unsuitable reference pictures, as this means that they are aware of what constitutes a poor placement and are therefore less likely to make these mistakes themselves.

This is slightly contradictory to the results on the use of the report function during the game. While few participants reported pictures, those who did, often overused this opportunity and reported more picture pairs than needed. In practice, it is tricky to decide where to set the limit between a suitable and unsuitable picture. For the majority of the reference pictures submitted via the CrowdWater app the staff gauge placements are neither perfect nor useless. Although many staff gauges are not placed ideally, this does not necessarily mean that they are unusable. Depending on the location, it is often also not possible to set the

virtual staff gauge perfectly.

There was no strong correlation between the game score and the improvements after the training. This could partly be due to the fact that the learning process might take place gradually during the game. Early in the game, participants might get few points and improve later during the game, leading to an average game score and a learning effect before finishing the training. The number of game rounds for optimal training is unknown but the four rounds used here may be a good compromise between

showing enough different pictures and not taking too much time. Strobl et al. (2019) showed that, on average, players who played more than two rounds of the game (24 picture pairs) chose the right water level class more often than players who played fewer rounds. Players who played more than four rounds (48 picture pairs) were even more accurate.

## 5.2 Advantages and disadvantages of using an online citizen science game for training

The primary goal of the CrowdWater game is quality control of the crowdsourced data by the citizen scientists themselves.

This method has proven successful in improving the water level class data (Strobl et al., 2019). The idea to use the game also for training developed over time (see Sect. 2.3). By using an online game for this dual purpose (quality control and training) less effort from project administrators is needed, compared to developing a separate online training module and quality control mechanism. Newman et al. (2010) developed multimedia tutorials for a species identification citizen science project and pointed out that they *"[...] found the development of multimedia tutorials difficult and time consuming."* (Newman et al.,

2010, p. 284).

The CrowdWater game goes beyond the separation of data quality control into *"training before the task"* and *"checking after the task"* (Freitag et al., 2016). Instead training and checking are combined in a continuous loop, where new citizen scientists





train and more experienced citizen scientists check the data with the same task. This in turn converts new citizen scientists into more experienced ones after only a few rounds of playing the game. This is similar to iSpot, where citizen scientists upload a

picture of a species and identify the species, which is then checked online by other contributors (Silvertown et al., 2015). This leads to the new citizen scientists learning more about species, which will in turn make them better at helping other citizen scientists in the future. The approach by Bonter and Cooper (2012) for the FeederWatch project also combined data quality control with training by sending an automatic message to the contributor when a rare and possibly unlikely entry was submitted. They state that these *"[...] messages may function as training tools by encouraging participants to become more

knowledgeable [...]"* (Bonter and Cooper, 2012, p. 306). However, the CrowdWater game is different from these projects in that it does not provide factual knowledge (e.g. on streams or hydrology).

The inclusion of new (and therefore unexperienced) citizen scientists in the quality control process did not negatively influence the quality of the data, mainly due to the averaging of votes of several players (Strobl et al., 2019). Of course this is only the case as long as there are still enough experienced players, playing the game as well. In the project iSpot, the issue of including

beginners in the validation process was solved through reputation scores, which need to be earned through correct species suggestions (Silvertown et al., 2015). This could also be a next step for the CrowdWater game, where an accuracy score can be calculated for each citizen scientist, which can then be used to weight water level class votes in the game to obtain the correct water level class (Strobl et al., 2019). However, the fact that four rounds of playing the game seem sufficient for training, suggests that this is not necessary because new game players quickly turn into experienced ones.

If a citizen science project wants to develop a training task (as opposed to a quality control methodology that also works as a training task), slightly different approaches might be better. In our case, pointing out the essential criteria for setting a staff gauge in between the picture pairs might have been helpful. Similarly, feedback about the correct water level class could be given after each picture pair, as opposed to after each round of the game (as it is currently implemented). However, this would likely disturb the frequent players and consequently our primary goal, data quality control, as most game players are already

aware of these criteria and do not want to be disrupted after every picture when they play the game. Therefore, we decided not to add this information to the game. However, additional material, such as tutorial videos, an app manual including *"good"* and *"bad"* staff gauge placement examples, and introductory app slides are available on the project homepage. However, our personal experience is, that many citizen scientists do not look at this material before using the app and are often not aware of it. A potential benefit of the game, compared to the theoretical material, is that citizen scientists are less likely to see it as

*"homework"* but more as an entertaining activity and are, therefore, likely to spend more time with the game than they would do with other information materials. Encouragingly, participants of this study overall enjoyed playing the game, meaning that they would participate for the fun aspect instead of seeing it as a *"learning task"*. Consequently, the game can be recommended to any potential citizen scientist, without first having to assess their skills, i.e. their need for training. Additionally, when trying not to demotivate new users, we can recommend to users that they play the game instead of discouraging them by explaining

that their observations are incorrect.

In the future, it might be feasible to require participants to play the game before starting a new water level class measurement location and thus placing a virtual staff gauge in the CrowdWater app. This would be easily verifiable, as the app and game accounts are the same. In contrast, it is difficult to assess if citizen scientists actually read through the introductory slides on the app or the theoretical training material that is offered online. Having a compulsory task before all features of the

CrowdWater app become available might heighten the entry barrier, which most citizen science projects that require many participants try to avoid. However, it could also be argued that participants, who chose to complete a training session, might be more committed towards a project and might therefore become more reliable long-term citizen scientists.



### 5.3 Does participants' self-assessment of confidence, difficulty and fun predict performance?

In general, participants were more confident about their performance and thought that the task was easier after the training.
Self-assessment, however, seems to be an unreliable proxy for actual performance and should therefore be interpreted carefully. Participants with a low score for placing or rating the virtual staff gauges might not have realised what the essential criteria were (hence the low score) and therefore also did not realise that their staff gauge placement or rating of the reference pictures was not ideal. Self-assessment might improve after a while, once participants are more aware of which criteria to look for. Such a realisation was seen by a CrowdWater app user, who commented on one of his own data submissions that new
observations were relatively difficult because the virtual staff gauge, which he had placed some months earlier, was not placed ideally. This indicates that the sequence of activities in the CrowdWater project is not ideal, as volunteers have to start with the most difficult part, without having been confronted with different staff gauge placement options. It also suggests that after a while citizen scientists learn what criteria to look out for and that training may be useful.

The predictability of performance based on self-assessment seems to vary for other studies. McDonough et al. (2017) found
that the self-assessed species identification skills did not correspond to the skills of the citizen scientists. Starr et al. (2014) identified a group of citizen scientists who seemed too confident in their abilities, but overall believed that the self-assessment of the majority of their citizen scientists was accurate. Crall et al. (2011) found that citizen scientists' skills increased with their self-assessed comfort level. Further research would be required to determine when self-assessment can be a reliable prediction of performance. In the meantime, self-assessments should not be fully relied on or used as a proxy for data quality.
Citizen science project tasks and therefore also training tasks should always be designed "*[...] with the skill of the citizens in mind [...]*" (Aceves-Bueno et al., 2017, p. 287). In this study a similar number of participants rated the game as easy, neutral or difficult. This gives the impression that the difficulty of the game is at a reasonably good level, as it is meant to be engaging and exciting but at the same time not too challenging to hinder participation. It should be noted that the participants in this study looked at 50 picture pairs in a row, to simulate several rounds of the regular CrowdWater game, which only shows
twelve picture pairs per day. The CrowdWater game itself is therefore likely even more accessible because it is less time consuming (and tiring) for citizen scientists.

### 5.4 Limitations of the study

The study was standardised by providing a number of pictures of the same stream to the participants to make the rating of their staff gauge placement more comparable and independent of their ability to find a suitable stream. We included a wide range
of stream pictures, including some unsuitable angles. It is encouraging to see that there was no difference in the performance of placing the staff gauge after the training online and outdoor, indicating that the online interface and the app interface were equally intuitive and that participants could also find suitable stream sections on their own. The training therefore seems to be teaching the necessary skills to the participants.

Participants could choose from the same 18 stream pictures before and after the training, which could potentially lead to a
confirmation bias, i.e. participants might be more likely to choose the same picture after the training as they did before the training. We believe that this effect was negligible, as only two participants with a poor choice of the stream picture before the training still had a poor score after the training as well. All other participants either changed the picture, or had already chosen a suitable picture before the training.

By singling out participants with a poor performance before the training, the natural variation in the performances might lead
to an improved performance after the training due to a regression towards the mean. However, the improvements were statistically significant when analysed for all participants as well. Further research should investigate how many rounds of the game would be optimal for training the average citizen scientist and if more rounds might train those participants who still received low scores after the training, i.e. if the optimal number of rounds could be adapted depending on the citizen scientist.





A disproportionate large number of study participants in this study had a university degree (85 %). The reasons were a bias in the social network of the authors, recruitment at the university, a tendency of people being more interested in university studies if they have been to university themselves, and the study being conducted in English. Many other citizen science projects also report a higher participation of university educated citizen scientists (Brossard et al., 2005; Crall et al., 2011; Overdevest et al., 2004), indicating that the participants of this study might not be that different from the actual citizen scientist in the CrowdWater project.

## 6 Conclusions

We investigated the value of an online game as a training tool for the CrowdWater project. This game was initially designed for data quality control but turned out to be valuable to improve the participants' ability to set-up new observation locations as well. Our results are encouraging beyond the CrowdWater project and we argue that the overall conclusions that: 1) games can provide a suitable approach for training and 2) training and data quality control can be combined also apply to other citizen science projects. Based on our study, the following conclusions about games for training in citizen science projects can be made:

- Citizen science projects should, if possible, be kept intuitive and easy, as this lowers the entry barrier and might prevent misunderstandings. For the placement of the staff gauge in the CrowdWater project, 70 % of the participants of this study already did well before receiving any training. This compares well, with the approximately 10 % error rate for data submitted through the app (Seibert et al., 2019a).

- Games facilitate training of new citizen scientists and people who have already participated for a while. A big advantage is that this approach is scalable. Large projects with a lot of beginners are also likely to have a lot of advanced citizen scientists and therefore the number of people who can be trained is not limited by the number of people managing the project.

- Training through a game might not necessarily be perceived as training by the citizen scientists (in our case the primary goal is data quality control). Potentially this helps to make the training feel less like *"homework"* before starting to collect data. Nearly two thirds of the participants of this training study said that the game was fun, this compares well with a survey among early game players of whom 86 % said that they enjoyed playing the game (Strobl et al., 2019).

- While materials such as manuals and tutorials can be useful, gamified approaches provide an enjoyable alternative training mechanisms for citizen scientists. Citizen scientists might respond differently to various training techniques. In our case we noticed that few citizen scientists read the manual or watch the instruction videos but also that some individuals might have responded better to a more explicit and less playful training method. We, therefore, recommend offering different training options.

## 7 Data availability

All data files are available from the Zenodo data repository (10.5281/zenodo.3538008).

## 8 Author contributions

Barbara Strobl: Conceptualization, Data curation, Formal analysis, Investigation, Methodology, Project administration, Resources, Visualization, Writing – original draft

Simon Etter: Conceptualization, Methodology, Writing – review & editing

Ilja van Meerveld: Conceptualization, Funding acquisition, Methodology, Supervision, Writing – review & editing

Jan Seibert: Conceptualization, Funding acquisition, Methodology, Supervision, Writing – review & editing





**9 Competing interests**

The authors declare that no competing interests exist.



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
