# Peer review of "Training citizen scientists through an online game developed for data quality control"

_Geoscience Communication, 2019_

## Referee Comment (RC1) · Anonymous Referee #1 · 6 Jan 2020

General Comments This is a well written paper that focus on using a training game to educate citizen scientists on how to measure stream stage using the CrowdWater system. The novelty of this paper is its focus on quantifying uncertainty in citizen science data pre- and post-training using a gamification approach. Results show improvements in the users with the lowest accuracy but in general results highlight most users (70%) do a very good job before training/gamification is provided. I know of no other paper in the hydrologic sciences that has taken the approach presented here and as a result believe this paper will be a good addition to the literature. As a result of CrowdWater being a smart phone application-based system the methods described here should be transferable to other such platforms. I see the future of citizen since being smart phone application based and this will lay a good groundwork for training.

[Figure]

Specific Comments

Is there any way to get information on the six people that did not complete all the tasks? Was the training to time intensive or were they just not interested in the project.

Good Staff Gauge Placement vs. Good Rating Score: It would be interesting to know if there is any correlation between how well a user places a gauge and how well a user can identify a good gauge location. If there is a positive correlation would it be possible to just use one of these two training/gaming methods? While there is a bit of a novelty in playing this game, I think that some users may have short attention and too much training will turn them off of the system. Is there any way to know what the minimum training would be needed in order to improve the 30% that were below the acceptable level?

It is unclear what the significance was of a good game score being set at 245. Was this a pre-determined metric or a natural split in our participation data?

Use of "app" in Figure 7: The third column of the box plot is labeled "app", which I believe represent the user actually placing their own gauge in the application after doing through the training. I would suggest changing this label to something that makes it clear this is in the field/outdoor use of the system. This is the real-world implementation of CrowdWater. Just labeling "app" does not show that these users are now "going live with the CrowdWater system".

Do the study participants continue to use of CrowdWater? If so do they contribute more than the "average" user. It may be too soon to tell but it would be interesting to know through the use of gamification if users are more engaged for longer periods of time.

Technical Corrections: Line 70: Suggest removing the word "study" to simplify the sentence. Line 80: Citation for Goodchild should be 2007 Line 168: Suggest using the terms Pre-Training and Post-Training. Line 387: Suggest removing the word "already" to simplify the sentence. Line 705: Remove Paul, J.D. from author list

Interactive
comment

---

## Author Comment (AC1) · 30 Jan 2020

***Interactive comment*** on "Training citizen scientists through an online game developed for data quality control" by Barbara Strobl et al.

**Anonymous Referee #1**

General Comments

This is a well written paper that focus on using a training game to educate citizen scientists on how to measure stream stage using the CrowdWater system. The novelty of this paper is its focus on quantifying uncertainty in citizen science data pre- and post-training using a gamification approach. Results show improvements in the users with the lowest accuracy but in general results highlight most users (70%) do a very good job before training/gamification is provided. I know of no other paper in the hydrologic sciences that has taken the approach presented here and as a result believe this paper will be a good addition to the literature. As a result of CrowdWater being a smart phone application-based system the methods described here should be transferable to other such platforms. I see the future of citizen since being smart phone application based and this will lay a good groundwork for training.

We thank the referee for her/his review and positive feedback. The individual comments are addressed below (our text in blue).

Specific Comments

Is there any way to get information on the six people that did not complete all the tasks? Was the training to time intensive or were they just not interested in the project.

The first author was in contact with most people who did not complete the last task (start a new measurement location with the app) and tried to remind them via email to do so. Most of them indicated that they had not had time to do it yet, but intended to do so (but then didn't). Only one person replied that he/she would not have time to complete the task. We assume that the problem was mainly related to remembering the task when they were close to a river, as not everyone lived near a river. We will add a brief explanation to the paper (Line 285) *"When sending email reminders to complete the task, most participants indicated a lack of time or a suitable nearby river. Most still intended to complete the task, but forgot about it."*

Good Staff Gauge Placement vs. Good Rating Score: It would be interesting to know if there is any correlation between how well a user places a gauge and how well a user can identify a good gauge location. If there is a positive correlation would it be possible to just use one of these two training/gaming methods? While there is a bit of a novelty in playing this game, I think that some users may have short attention and too much training will turn them off of the system. Is there any way to know what the minimum training would be needed in order to improve the 30% that were below the acceptable level?

There might be a bit of a misunderstanding here. The staff gauge placement and rating tasks were merely used as tests to see if the performance of the participants improved by playing the game. The training itself thus only consists of playing the CrowdWater game and only this task is recommended for new citizen scientists. We agree that it is important to minimize the effort for new citizen scientists. Unfortunately we do not know what the minimum training is as only one version of the training (i.e., 50 picture pairs) was tested and therefore we do not have the data to answer this question. We will carefully read through the document and rephrase sentences that might have caused this confusion.

Out of curiosity, we looked into the correlation between the staff gauge rating (before and after training) and the game score. It was not significant (r = 0.02 and 0.14 respectively). This lack of correlation might be partly due to the learning process while playing the game. In the beginning some people may get a low score when playing the game and improve throughout. This yields an overall average game score, but the learning process helped them improve their rating score after the training.

It is unclear what the significance was of a good game score being set at 245. Was this a pre-determined metric or a natural split in our participation data?

This was a pre-determined metric that reflects a situation where a participant chooses the correct class for 35 out of the 50 picture pairs, was one class off five times, more than one class off for another five picture pairs, and reported five pictures (we considered five pictures unsuitable and would thus have reported them). We will adjust the text to clarify that this was a pre-determined threshold: *"The threshold for a good game score was set pre-determined at 245 points …"* (Line 256). There was no natural gap of scores around this threshold, that would have facilitated a split after the survey. There is a small gap around the median (as can be seen in Figure 4), however, we don't think that this reflects a real gap but is rather a coincidence due to the limited number of data points.

Use of "app" in Figure 7: The third column of the box plot is labeled "app", which I believe represent the user actually placing their own gauge in the application after doing through the training. I would suggest changing this label to something that makes it clear this is in the field/outdoor use of the system. This is the real-world implementation of CrowdWater. Just labeling "app" does not show that these users are now "going live with the CrowdWater system".

Thank you for this comment. We agree and will adjust the label to "outdoor use" to better reflect the task.

Do the study participants continue to use of CrowdWater? If so do they contribute more than the "average" user. It may be too soon to tell but it would be interesting to know through the use of gamification if users are more engaged for longer periods of time.

We agree that this would have been interesting information. Unfortunately we cannot verify this. The participants were given a username for this exercise. To later use the app, they had to create their own account with a different username. We have no way to link the two usernames. We are aware of other CrowdWater citizen scientists, who started with the game and later used the app, but unfortunately have no numbers to quantify this. We conducted a survey with CrowdWater game players for a different publication (Strobl et al., 2019). In this survey, 67 % of the respondents had also used the CrowdWater app, however 79 % of respondents who had used both had started with the CrowdWater app. At the time the survey closed (February 2019) the CrowdWater app had been available for almost two years, whereas the CrowdWater game had only been public for a bit more than half a year. It is therefore possible that the results might be different if we would conduct the same survey now.

Technical Corrections:

Line 70: Suggest removing the word "study" to simplify the sentence.

Thank you for this suggestion, we will adapt the manuscript accordingly.

Line 80: Citation for Goodchild should be 2007

Sorry, but we did not find this error. The reference is mentioned with the year 2007 in the text and in the references. Please let us know if we missed something.

Line 168: Suggest using the terms Pre-Training and Post-Training.

We will adapt the text accordingly.

Line 387: Suggest removing the word "already" to simplify the sentence.

We will remove the word "already".

Line 705: Remove Paul, J.D. from author list

Thank you for making us aware of this mistake, we will correct the reference.

References:

Strobl, B., Etter, S., van Meerveld, I. and Seibert, J.: The CrowdWater game: A playful way to improve the accuracy of crowdsourced water level class data, edited by S. Mirjalili, PLoS One, 14(9), e0222579, doi:https://doi.org/10.1371/journal.pone.0222579, 2019.

---

## Referee Comment (RC2) · Anonymous Referee #2 · 4 Feb 2020

This is a well written and interesting paper on testing how well an online game can be used to improve the quality of the data collected through the CrowdWater app. The paper is well grounded in the existing literature and critically assesses the results against those found in other studies, sometimes contradictory and sometimes confirmatory. I thought the experiment was set up very well and the tests used to examine different aspects of quality as well as self-assessment were appropriate and well-interpreted. This paper provides a high quality contribution to the citizen science/hydrology literature.

My only real comment is about the demographic data that were collected. It would be interesting to see whether this could be used in some type of decision tree to then target the type of training you present to your participants or to decide on the amount of pictures that should be shown to people. For example, what is the background/age

of the people who found the game difficult? Did you consider asking more information about past experience with citizen science, online games, mobile apps, etc.? You might then be able to fit a model where quality is a function of different demographic variables. This would tell you if any of them are significant and whether an increase/decrease in a variable results in an increase/decrease in performance. You might need a larger data set for this but it could be an interesting future study.

A very minor comment is that there are a few minor errors in the writing. These can be picked up by a thorough read/edit of the paper.

---

## Author Comment (AC2) · 7 Feb 2020

*Interactive comment* on "Training citizen scientists through an online game developed for data quality control" by Barbara Strobl et al.

**Anonymous Referee #2**

This is a well written and interesting paper on testing how well an online game can be used to improve the quality of the data collected through the CrowdWater app. The paper is well grounded in the existing literature and critically assesses the results against those found in other studies, sometimes contradictory and sometimes confirmatory.  I thought the experiment was set up very well and the tests used to examine different aspects of quality as well as self-assessment were appropriate and well-interpreted. This paper provides a high quality contribution to the citizen science/hydrology literature.

We thank the referee for these kind words and her/his interesting suggestions. The individual comments are addressed below (our text in blue).

My only real comment is about the demographic data that were collected. It would be interesting to see whether this could be used in some type of decision tree to then target the type of training you present to your participants or to decide on the amount of pictures that should be shown to people. For example, what is the background/age of the people who found the game difficult? Did you consider asking more information about past experience with citizen science, online games, mobile apps, etc.? You might then be able to fit a model where quality is a function of different demographic variables. This would tell you if any of them are significant and whether an increase/decrease in a variable results in an increase/decrease in performance. You might need a larger dataset for this but it could be an interesting future study.

We agree that a question regarding previous experience with citizen science, gaming and smartphones would have been beneficial. Unfortunately, we did not ask for this information at the time of the survey. All we know is that the participants had not used the CrowdWater app, nor played the CrowdWater game before. We don't have any information about their familiarity with other games or apps. Unfortunately, the numbers are too small to draw any robust conclusions regarding the importance of demographic variables on the effectiveness of the training (e.g., to divide the participants that did not perform well in the pre-training tests by age).

We will add a sentence to the manuscript highlighting this for potential future research. *"In future studies the effect of previous experiences with citizen science, online games or smartphones in general could be investigated. This would provide an indication of who might require more training or for whom training via a game is most beneficial."* (L 495)

A very minor comment is that there are a few minor errors in the writing. These can be picked up by a thorough read/edit of the paper.

We will carefully proof-read the manuscript before resubmitting it.

---

## Author Response (AR1)

General Comments

This is a well written paper that focus on using a training game to educate citizen scientists on how to measure stream stage using the CrowdWater system. The novelty of this paper is its focus on quantifying uncertainty in citizen science data pre- and post-training using a gamification approach. Results show improvements in the users with the lowest accuracy but in general results highlight most users (70%) do a very good job before training/gamification is provided. I know of no other paper in the hydrologic sciences that has taken the approach presented here and as a result believe this paper will be a good addition to the literature. As a result of CrowdWater being a smart phone application-based system the methods described here should be transferable to other such platforms. I see the future of citizen since being smart phone application based and this will lay a good groundwork for training.

We thank the referee for her/his valuable review comments and overall positive assessment of our study. The individual comments are addressed below (our text in blue).

Specific Comments

Is there any way to get information on the six people that did not complete all the tasks? Was the training to time intensive or were they just not interested in the project.

The first author was in contact with most people who did not complete the last task (i.e., start a new measurement location with the app) and tried to remind them via email to do so. Most of them indicated that they had not had time to do it yet, but intended to do so (but then did not do so). Only one person replied that he/she would not have time to complete the task. We assume that the problem was mainly related to remembering the task when they were close to a river, as not everyone lives near a river. We added a brief explanation to the paper (Line 285) *"When sending email reminders to complete this sixth task, several participants indicated a lack of time or a suitable nearby river. Most participants intended to complete the task, but forgot about it in the end."*

Good Staff Gauge Placement vs. Good Rating Score: It would be interesting to know if there is any correlation between how well a user places a gauge and how well a user can identify a good gauge location. If there is a positive correlation would it be possible to just use one of these two training/gaming methods? While there is a bit of a novelty in playing this game, I think that some users may have short attention and too much training will turn them off of the system. Is there any way to know what the minimum training would be needed in order to improve the 30% that were below the acceptable level?

There might be a bit of a misunderstanding here. The staff gauge placement and rating tasks were merely used as tests to see if the performance of the participants improved by playing the game. The training itself thus only consists of playing the CrowdWater game, and only this task is recommended for new citizen scientists. We agree that it is important to minimize the effort for new citizen scientists. Unfortunately, we do not know what the minimum training is as only one version of the training (i.e., 50 picture pairs) was tested and therefore we do not have the data to answer this question. We have added a sentence to clarify this point (Line 169) *"Pre-training and post-training tasks are only intended to assess the participant's performance during this study and are not part of the training for the CrowdWater project."*. We also highlight on Line 509 that we did not investigate the optimal number of rounds of the game that need to be played for training and that this requires further study.

Out of curiosity, and in response to the reviewer's comment, we looked into the correlation between the staff gauge rating (before and after training) and the game score. However, this correlation was not significant (r = 0.02 and 0.14 respectively). This lack of correlation might be partly due to the learning process while playing the game. In the beginning, some people may get a low score when playing the game but improve throughout the game, resulting in an overall average game score, even though the learning process helped them to improve their rating score after the training (see also Line 507).

It is unclear what the significance was of a good game score being set at 245. Was this a pre-determined metric or a natural split in our participation data?

This was a pre-determined metric that reflects a situation where a participant chooses the correct class for 35 out of the 50 picture pairs,. For the remaining 15 picture pairs, they could be one class off five times, more than one class off for another five picture pairs, and reported five pictures (we considered five pictures unsuitable and would thus have reported them). We adjusted the text to clarify that this was a pre-determined threshold: *"The threshold for a good game score was determined before the study and set at 245 points …"* (Line 256). There was no natural gap of scores around this threshold that would have facilitated a split after the survey. There is a small gap around the median (as can be seen in Figure 4). However, we don't think that this reflects a real difference but is rather a coincidence that can be explained by the limited number of data points.

Use of "app" in Figure 7: The third column of the box plot is labeled "app", which I believe represent the user actually placing their own gauge in the application after doing through the training. I would suggest changing this label to something that makes it clear this is in the field/outdoor use of the system. This is the real-world implementation of CrowdWater. Just labeling "app" does not show that these users are now "going live with the CrowdWater system".

Thank you for this comment. We agree and adjusted the label to "outdoor use" to better reflect the task.

Do the study participants continue to use of CrowdWater? If so do they contribute more than the "average" user. It may be too soon to tell but it would be interesting to know through the use of gamification if users are more engaged for longer periods of time.

We agree that this would have been interesting information. Unfortunately, we cannot verify this. The participants were given a username for this exercise. To later use the app, they had to create their own account with a different username. We have no way to link the two usernames. We are aware of other CrowdWater citizen scientists, who started with the game and later decided also to use the app, but unfortunately have no numbers to quantify this. We conducted a survey with CrowdWater game players for a different publication (Strobl et al., 2019). In this survey, 67 % of the respondents had also used the CrowdWater app, however, 79 % of respondents who had used both game and app, had started with using the CrowdWater app. At the time the survey closed (February 2019) the CrowdWater app had been available for almost two years, whereas the CrowdWater game had only been public for a bit more than half a year. It is, therefore likely that the results would be different if we conduct the survey again.

Technical Corrections:

Line 70: Suggest removing the word "study" to simplify the sentence.

Thank you for this suggestion. We adapted the manuscript accordingly.

Line 80: Citation for Goodchild should be 2007

Sorry, but we did not find this error. The reference is mentioned with the year 2007 in the text and in the reference list. Please let us know if we missed something.

Line 168: Suggest using the terms Pre-Training and Post-Training.

We adapted the text accordingly.

Line 387: Suggest removing the word "already" to simplify the sentence.

We removed the word "already".

Line 705: Remove Paul, J.D. from author list

Thank you for making us aware of this mistake. We corrected the reference.

References:

Strobl, B., Etter, S., van Meerveld, I. and Seibert, J.: The CrowdWater game: A playful way to improve the accuracy of crowdsourced water level class data, PLoS One, 14(9), e0222579, https://doi.org/10.1371/journal.pone.0222579, 2019.

**Anonymous Referee #2**

This is a well written and interesting paper on testing how well an online game can be used to improve the quality of the data collected through the CrowdWater app. The paper is well grounded in the existing literature and critically assesses the results against those found in other studies, sometimes contradictory and sometimes confirmatory. I thought the experiment was set up very well and the tests used to examine different aspects of quality as well as self-assessment were appropriate and well-interpreted. This paper provides a high quality contribution to the citizen science/hydrology literature.

We thank the referee for these kind words and her/his valuable suggestions. The individual comments are addressed below (our text in blue).

My only real comment is about the demographic data that were collected. It would be interesting to see whether this could be used in some type of decision tree to then target the type of training you present to your participants or to decide on the amount of pictures that should be shown to people. For example, what is the background/age of the people who found the game difficult? Did you consider asking more information about past experience with citizen science, online games, mobile apps, etc.? You might then be able to fit a model where quality is a function of different demographic variables. This would tell you if any of them are significant and whether an increase/decrease in a variable results in an increase/decrease in performance. You might need a larger dataset for this but it could be an interesting future study.

We agree that a question regarding previous experience with citizen science, gaming and smartphones would have been beneficial. Unfortunately, we did not ask for this information at the time of the survey. We only know that the participants had not used the CrowdWater app, nor played the CrowdWater game before. We don't have any information about their familiarity with other games or apps. Unfortunately, the numbers are too small to draw any robust conclusions regarding the importance of other demographic variables on the effectiveness of the training (e.g., to divide the participants that did not perform well in the pre-training tests by age).

We added a sentence to the manuscript highlighting this for potential future research. *"We do not know if the participants who benefited most from playing the game had previous experiences with citizen science, online games or smartphones. This could be investigated in a future study and would indicate who might require more training or for whom training via a game is most beneficial."* (L 495)

A very minor comment is that there are a few minor errors in the writing. These can be picked up by a thorough read/edit of the paper.

We carefully proof-read the manuscript before resubmission.

[revised manuscript text omitted]

---

## Author Response (AR2)

We thank the editor for the technical corrections.

We have adapted the last comment slightly. The adapted section now reads:

[revised manuscript text omitted]